# *In vitro* activity of ceftazidime/avibactam, cefiderocol, meropenem/vaborbactam and imipenem/relebactam against clinical strains of the *Stenotrophomonas maltophilia* complex

Braulio Josué Méndez-Sotelo[1☯], Mónica Delgado-Beltrán[2☯], Melissa Hernández-Durán[3], Claudia Adriana Colín-Castro[3], José Esquivel-Bautista[4], Sandra Angélica Ortega-Oliva[4], Jossue Ortiz-Álvarez[5], Rodolfo García-Contreras[6], Rafael Franco-Cendejas[7]*, Luis Esau Lopez Jacome[3,8]*

1 Infectious Diseases Division, Instituto Nacional de Rehabilitación Luis Guillermo Ibarra Ibarra, Mexico City, Mexico, 2 Infectious Diseases Department, Instituto Nacional de Cancerología, Mexico City, Mexico, 3 Clinical Microbiology Laboratory, Infectious Diseases Division, Instituto Nacional de Rehabilitación Luis Guillermo Ibarra Ibarra, Mexico City, Mexico, 4 Centro Nacional de Referencia de Inocuidad y Bioseguridad Agroalimentaria, Servicio Nacional de Sanidad, Inocuidad y Calidad Agroalimentaria (SENASICA), Tecámac, Mexico State, Mexico, 5 Programa "Investigadoras e Investigadores por México", Consejo Nacional de Humanidades, Ciencias y Tecnologías (CONAHCYT), Mexico City, Mexico, 6 Medicine Faculty, Bacteriology Laboratory, Microbiology and Parasitology Department, Universidad Nacional Autónoma de México, Mexico City, Mexico, 7 Instituto Nacional de Rehabilitación Luis Guillermo Ibarra Ibarra, Biomedical Research Subdirection, Mexico City, Mexico, 8 Chemistry Faculty, Biology Department, Universidad Nacional Autónoma de México, Mexico City, Mexico

☯ These authors contributed equally to this work.
* esaulopezjacome@quimica.unam.mx (LELJ); raffcend@yahoo.com (RF-C)

## Abstract

### Background

Infections caused by *Stenotrophomonas maltophilia* and related species are increasing worldwide. Unfortunately, treatment options are limited, whereas the antimicrobial resistance is increasing.

### Methods

We included clinical isolates identified as *S. maltophilia* by VITEK 2 Compact. Ceftazidime/avibactam, meropenem/vaborbactam, imipenem/relebactam, cefiderocol, quinolones, and tetracycline family members were evaluated by broth microdilution method and compared with first-line treatment drugs. Minimum inhibitory concentrations (MICs) were reported for all antibiotics. We sequenced the Whole Genome of cefiderocol resistant strains (CRSs) and annotated their genes associated with cefiderocol resistance (GACR). Presumptive phylogenetic identification employing the 16S marker was performed.

### Results

One hundred and one clinical strains were evaluated, sulfamethoxazole and trimethoprim, levofloxacin and minocycline showed susceptibilities of 99.01%, 95.04% and 100%

**Data Availability Statement:** All relevant data are within the paper. The Whole Genome Shotgun project has been deposited at DDBJ/ENA/GenBank under the accession numbers JAWJTG000000000, JAWJTH000000000, JAWJTI000000000, JAWJTJ000000000 and JAWJTK000000000. The version described in this article are available under accession numbers JAWJTG010000000, JAWJTH010000000, JAWJTI010000000, JAWJTJ010000000 and JAWJTK010000000.

**Funding:** The author(s) received no specific funding for this work.

**Competing interests:** The authors have declared that no competing interests exist.

respectively. Ceftazidime was the antibiotic with the highest percentage of resistance in all samples (77.22%). Five strains were resistant to cefiderocol exhibiting MIC values $\geq 2$ µg/mL (4.95%). The β-lactamase inhibitors meropenem/vaborbactam and imipenem/relebactam, failed to inhibit *S. maltophilia*, preserving both $MIC_{50}$ and $MIC_{90} \geq 64$ µg/mL. Ceftazidime/avibactam restored the activity of ceftazidime decreasing the MIC range. Tigecycline had the lowest MIC range, $MIC_{50}$ and $MIC_{90}$. Phylogeny based on 16S rRNA allowed to identify to cefiderocol resistant strains as putative species clustered into *Stenotrophomonas maltophilia* complex (Smc). In these strains, we detected GARCs such as Mutiple Drug Resistance (MDR) efflux pumps, L1-type β-lactamases, iron transporters and type-1 fimbriae.

## Conclusion

Antimicrobial resistance to first-line treatment is low. The *in vitro* activity of new β-lactamase inhibitors against *S. maltophilia* is poor, but avibactam may be a potential option. Cefiderocol could be considered as a potential new option for multidrug resistant infections. Tetracyclines had the best *in vitro* activity of all antibiotics evaluated.

## Introduction

*Stenotrophomonas maltophilia* is an aerobic, Gram-negative, non-fermenting rod capable of oxidizing glucose and maltose. It has a remarkable ability to adhere and form biofilms, allowing it to survive in a variety of aqueous environments, both natural and artificial settings. This Gram-negative rod is persistent in medical devices such as intravascular catheters, implantable devices, and mechanical ventilation circuits [1]. Previously considered as a microorganism of low virulence, it is now recognized as an opportunistic pathogen that can affect immunocompromised patients in hospital settings [2]. In addition, recent studies have demonstrated its pathogenic potential in immunocompetent individuals as successful [3]. *S. maltophilia* is clustered with other *Stenotrophomonas* genomospecies that share elevated levels of identity in 16S gene sequence conservation, forming the *Stenotrophomonas maltophilia* complex (Smc) [4,5].

Risk factors associated with invasive *S. maltophilia* infection include prolonged hospitalization in the intensive care unit, invasive procedures, mechanical ventilation, vascular access, prolonged exposure to antibiotics or corticosteroids, immunosuppression and solid organ or hematopoietic cell transplantation. In multivariate analyses, prolonged use of antibiotics (mainly carbapenems) was reported to be one of the major risk factors for developing *S. maltophilia* infections [6,7].

*S. maltophilia* is intrinsically resistant to several antibiotics, representing challenge for treatment. The antimicrobial resistance mechanisms described involve a combination of resistance mediated by plasmids [8], integrons [9], antibiotic-modifying enzymes, efflux pumps, and decreased membrane permeability [10,11]. The most reported [12] is the expression of two inducible chromosomal ß-lactamases L1 (a class B3 metallo-ß-lactamase) inhibiting carbapenems and L2 (class A cephalosporinase), which confers resistance to broad-spectrum cephalosporins and aztreonam, and can also be inactivated by clavulanic acid and avibactam [12].

Infections caused by *S. maltophilia* are increasing worldwide, the Antimicrobial Surveillance Program SENTRY study from 1997–2016 reported that it was the 6th cause of hospital-acquired pneumonia in intensive care units in the United States; in addition, in Latin America,

of the total isolates 56.8% belong to bacteremia, 34.8% to hospital-acquired pneumonia, and 6.8% to skin and soft tissue infections, with a mortality of 14–69% when an inadequate antibiotic was administered [13].

There are limited treatment options for infections caused by *S. maltophilia*, and CLSI (The Clinical & Laboratory Standards Institute) susceptibility breakpoints have only been described for ticarcillin/clavulanate (TIM), ceftazidime (CAZ), cefiderocol (FDC), minocycline (MIN), levofloxacin (LVX), trimethoprim-sulfamethoxazole (SXT) and chloramphenicol (CHL) [14], and for SXT and FDC in EUCAST (The European Committee on Antimicrobial Susceptibility Testing) [15]. The first-line treatment option remains SXT, which has an estimated susceptibility of 79–96% [10]. The SENTRY program reported that from 2009 to 2012, 96.3% of patients with *S. maltophilia* pneumonia had adequate susceptibility to SXT [16], although there are concerns about the increase of resistance in certain regions and the few alternatives in case of resistance [17].

CAZ and TIM are β-lactam antibiotics with significant activity against *S. maltophilia*. However, recent studies have shown a worrying increase in resistance rates of more than 30% for these antibiotics [18]. In comparison, the fluoroquinolones have shown a more favorable susceptibility profile when compared to CAZ and TIM, making them a suitable alternative treatment for SXT. However, it is important to note that an increase in resistance to LVX has been reported in some countries, particularly in the Asia-Pacific regions, reaching up to 30% [19]. Doxycycline (DO), tigecycline (TGC), and MIN have consistently shown good activity against *S. maltophilia* in numerous studies conducted over different time periods, sample types, and geographic regions. These antibiotics have shown susceptibility rates of up to 96% and can be considered viable treatment options for *S. maltophilia* infections [20].

The development of new β-lactamase inhibitors has opened the possibility of investigating new treatments against several microorganisms, such as *S. maltophilia*. Some studies with the new inhibitors showed that the addition of aztreonam (ATM) to CAZ with avibactam (AVI) improved *in vitro* activity from 82 to 97% [21], while Biagi et al published in 2020, AVI restored ATM susceptibility to 98%, compared to 61%, 71%, and 15% with clavulanate, relebactam, and vaborbactam, respectively [18].

FCD, a novel antibiotic consisting of cephalosporins linked to siderophores, has attracted considerable interest because of multidrug-resistant Gram-negative infections. One of its main advantages is its stability against both serine and metallo β-lactamases, making it a promising option in the fight against antibiotic resistance [22]. *In vitro*, studies have demonstrated high susceptibility rates of FDC against *S. maltophilia*, with some reports indicating almost 100% susceptibility [23]. However, clinical experience with FDC for *S. maltophilia* infections remains limited and is based primarily on experimental animal models.

Our aim was to describe the minimum inhibitory concentrations (MICs) of first and second-line antibiotics, as well as the new β-lactamase inhibitors, against invasive *S. maltophilia* isolates. By determining the MIC values, we aimed to provide valuable insights into the potential efficacy of these antimicrobial agents against *S. maltophilia*, thus helping to guide appropriate treatment strategies for infections caused by this pathogen.

## Results

### Clinical strains

A total of 219 confirmed clinical strains of *S. maltophilia* were collected during the study period. Of these, 118 (53.8%) were excluded from the analysis for various reasons: two were environmental samples, 32 were swab samples, 14 strains were recovered from catheter tips, one from sonication of prosthetic joint material, four had missing information in electronic

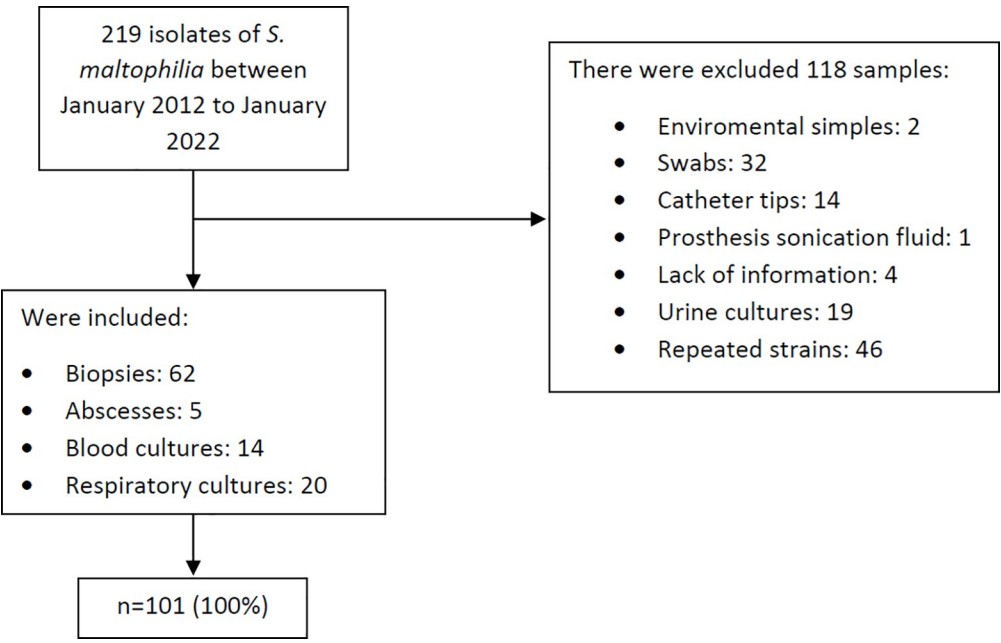

**Fig 1. Isolates selection algorithm.**

medical records, 19 were from urine cultures, and 46 were repeated isolates. As a result, we included 101 clinical specimens (Fig 1 and S1 Table).

## Susceptibility patterns

The isolates showed high susceptibility rates of 99.01%, 95.04%, and 100% to SXT, LVX, and MIN, respectively. There were no differences in SXT breakpoints between CLSI and EUCAST, as only one strain was found to be resistant according to both systems (also resistant to CAZ). During the period of 2015–2016, we identified 3 LVX-resistant isolates (3%) and 2 isolates with intermediate susceptibility (2%). Notably, all these LVX-resistant isolates were also found to be resistant to CAZ. Among all samples assessed, CAZ exhibited the highest percentage of resistance, with a rate of 77.22% (Table 1).

**Table 1. *In vitro* activities of selected agents against clinical *S. maltophilia* isolates.**

| Antibiotics | MIC (µg/mL) | | | | | | | | | | |
|---|---|---|---|---|---|---|---|---|---|---|---|
| | 0.06 | 125 | 0.25 | 0.5 | 1 | 2 | 4 | 8 | 16 | 32 | ≥64 |
| CAZ | 0 | 0 | 0 | 0 | 0 | 0 | 5 | 6 | 12 | 18 | 60 |
| SXT | 7 | 15 | 26 | 39 | 11 | 2 | 0 | 1 | 0 | 0 | 0 |
| LVX | 0 | 3 | 12 | 16 | 48 | 17 | 2 | 3 | 0 | 0 | 0 |
| MIN | 3 | 27 | 32 | 25 | 7 | 7 | 0 | 0 | 0 | 0 | 0 |
| FDC | 2 | 22 | 34 | 26 | 12 | 2 | 3 | 0 | 0 | 0 | 0 |
| MEV | 0 | 0 | 0 | 1 | 0 | 0 | 0 | 0 | 0 | 0 | 100 |
| IMR | 0 | 0 | 0 | 0 | 0 | 0 | 0 | 0 | 0 | 1 | 100 |
| CZA | 0 | 0 | 0 | 0 | 0 | 1 | 8 | 14 | 19 | 19 | 40 |
| TGC | 0 | 1 | 13 | 53 | 22 | 7 | 5 | 0 | 0 | 0 | 0 |

MIC, minimal inhibitory concentration; CAZ, ceftazidime; SXT, trimethoprim-sulfamethoxazole; LVX, levofloxacin; MIN, minocycline; FDC, cefiderocol; MEV, meropenem-vaborbactam; IMR, imipenem-relebactam; CZA, ceftazidime-avibactam; TGC, tigecycline.

**Table 2. MIC$_{50}$, MIC$_{90}$, ranges of MIC and susceptibilities of selected agents for *S. maltophilia* isolates from clinical specimens.**

| Antibiotics | MIC (µg/mL) | | | Susceptibility[a] n = 101 (%) | | |
|---|---|---|---|---|---|---|
| | Range | MIC$_{50}$ | MIC$_{90}$ | S | I | R |
| SXT | 0.06–8 | 0.5 | 1 | 100 (99.01) | - | 1 (0.99) |
| MIN | 0.06–2 | 0.25 | 1 | 101 (100) | 0 | 0 |
| CAZ | 4–64 | 32 | 64 | 11 (10.8) | 12 (12) | 78 (77) |
| LVX | 0.125–8 | 1 | 2 | 96 (95) | 2 (1.98) | 3 (2.97) |
| FDC | 0.06–4 | 0.25 | 1 | 96 (95) | - | 5 (4.95) |
| MEV | 0.5–64 | 64 | 64 | NA | NA | NA |
| IMR | 32–64 | 64 | 64 | NA | NA | NA |
| CZA | 2–64 | 16 | 64 | NA | NA | NA |
| TGC | 0.125–1 | 0.5 | 1 | NA | NA | NA |

MIC, minimal inhibitory concentration; S, susceptible; I, intermediate; R, resistant; NA, not applicable.

CAZ, ceftazidime; SXT, trimethoprim-sulfamethoxazole; LVX, levofloxacin; MIN, minocycline; FDC, cefiderocol; MEV, meropenem-vaborbactam; IMR, imipenem-relebactam; CZA, ceftazidime-avibactam; TGC, tigecycline.

[a]Susceptibilities of the isolates to the antimicrobial agents tested were determined according to the MIC interpretive breakpoints recommended by the CLSI.

The MIC, MIC$_{50}$ and MIC$_{90}$ ranges of antibiotics that did not have CLSI approved susceptibility breakpoints are described in Table 2. Both meropenem/vaborbactam (MEV) and imipenem/relebactam (IMR) showed no inhibitory activity against *S. maltophilia*, as indicated by their high MIC$_{50}$ and MIC$_{90}$values, both ≥64 µg/mL. However, a single strain showed low MIC values for these antibiotics (Table 2).

In the case of ceftazidime (CAZ), the addition of avibactam (CZA) resulted in an overall decrease in MIC$_{50}$ and MIC$_{90}$ compared to CAZ alone. Specifically, CAZ exhibited MIC$_{50}$ = 32 µg/mL and MIC$_{90}$ = 64 µg/mL, whereas CZA exhibited MIC$_{50}$ = 16 µg/mL and MIC90 = 64 µg/mL, indicating a significant reduction (p<0.001). Resemble to MIN, tigecycline (TGC) had the lowest range of MIC values, with both MIC$_{50}$ and MIC$_{90}$ falling within a favorable range (Table 2).

The annual susceptibility pattern for first-line antibiotics in the treatment of *S. maltophilia* infection SXT, LVX and CAZ is shown in Fig 2. It can be noted that LVX and SXT have

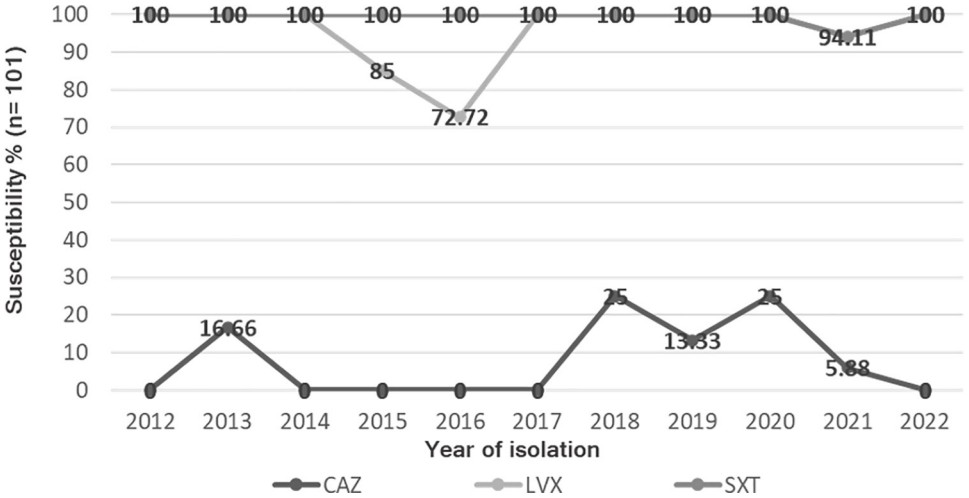

**Fig 2. Percentage of susceptibility per year of *S. maltophilia* to ceftazidime, levofloxacin, and trimethoprim-sulfamethoxazole.**

**Table 3.** *In vitro* activities to other antibiotics of cefiderocol resistant *S. maltophilia* isolates.

| Isolate | Susceptibility[a] | | | | MIC (µg/mL) | | | |
|---|---|---|---|---|---|---|---|---|
| | SXT | LVX | MIN | CAZ | CZA | MEV | IMR | TGC |
| **Isolate 1** | S | S | S | R | >64 | 64 | 64 | 0.5 |
| **Isolate 2** | S | S | S | R | 16 | 64 | 64 | 1 |
| **Isolate 3** | S | S | S | R | 64 | 64 | 64 | 0.5 |
| **Isolate 4** | S | S | S | R | 64 | 64 | 64 | 0.5 |
| **Isolate 5** | R | S | S | R | 64 | 64 | 64 | 4 |

MIC, minimal inhibitory concentration; S, susceptible; I, intermediate; R, resistant.

[a]Susceptibilities were determined according to the MIC interpretive breakpoints recommended by the CLSI. CAZ, ceftazidime; SXT, trimethoprim-sulfamethoxazole; LVX, levofloxacin; MIN, minocycline; MEV, meropenem-vaborbactam; IMR, imipenem-relebactam; CZA, ceftazidime-avibactam; TGC, tigecycline.

maintained their susceptibility pattern above 73% in the last 10 years, while CAZ has never reached a susceptibility higher than 25%.

### Cefiderocol-Resistant *Stenotrophomonas* strains

Finally, five samples defined as CRSS with MIC for FDC $\geq$ 2 µg/mL; of these strains, all were susceptible to LVX and MIN; also had a MIC lower than 4 µg/mL to TGC. We found only one CRSS resistant to SXT. One CRSS had a MIC for CZA of 16/4 µg/mL, the other four CRSS had a MIC $\geq$ 64 µg/mL. All CRSS that were resistant to CAZ with a MIC $\geq$ 16 µg/mL showed MIC $\geq$ 16µg/mL for CZA, MEV and IMR as well (Table 3).

A total of 14 GACRs were detected in the partial genome assembly of the five SRSS (Fig 3). The heat map plot shows a heterogeneous inclusion of GACRs among the SRSS. All GACRs were identified in the strain C960, while strain C1657 showed the lowest level of AREs. The transporters *tolQ*, *tonB* and *smeT* involved in the regulation of the multidrug resistance (MDR) efflux pump were found in all strains, as well as the core metabolic proteins *cysk1* and *panD*. Also, two components belonging to MDR (*smeE* and *smeF*) were detected in the five SRSS, but *smeD* was absent in C960 and C2852 but present in the rest of the strains. The β-lactamase L1 was identified in four strains but not in C1657.

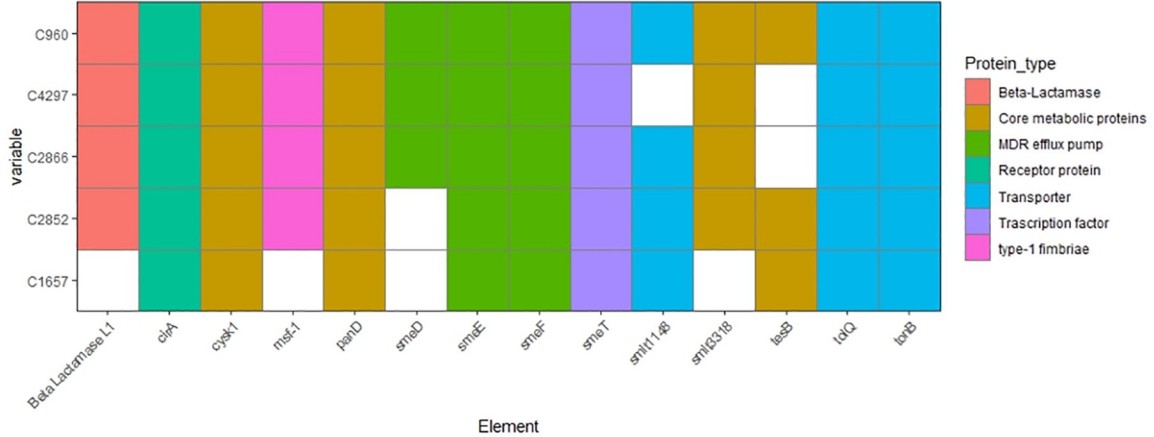

**Fig 3. Presence/absence plot of Genes Associated to Cefiderocol Resistance (GACR) detected in the genome of Cefiderocol-Resistant *Stenotrophomonas* Strains (CRSS).** The panel on the right rise represents the functional type of the GACR. Legends on the x-axis represent the proteins annotated. Legends of the y-axis represents the strain ID. Colored boxes represented the GACR detected in the strains. White boxes represent the GACR absents.

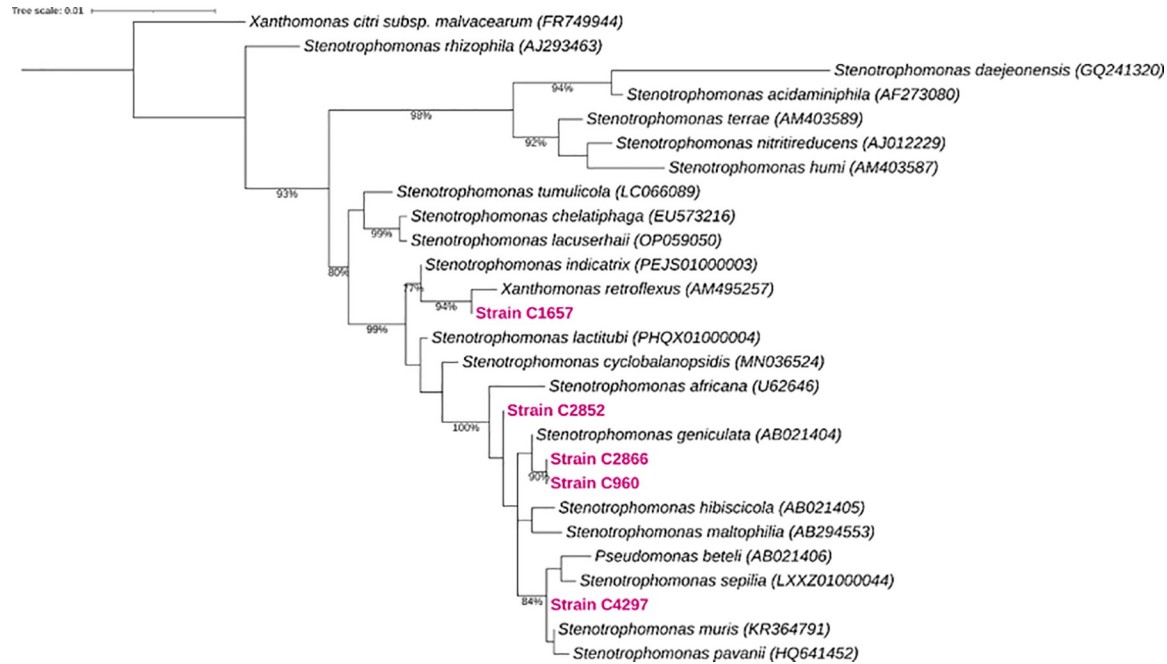

**Fig 4. ML phylogenetic tree of the 16S rRNA sequences.** The ID of the sequences from type strains are displayed into the brackets. The CRSS position in marked in bold and pink color into the tree. Numbers on the nodes and branches represent the Ultrafast Bootstrap values of 2000 replicates. Scale bar represents the number of nucleotide differences between branches.

Phylogenetic reconstructions showed that all SRSS were clustered into the Smc. Pairwise comparison of the 16S gene showed that C960, C2852, C2866 and C4267 were related to *Stenotrophomonas septilia* and *Stenotrophomonas geniculata* (99.2%), but C1657 showed <90% identity similarity (S1 Fig). However, the phylogeny and the pairwise comparison showed discordances among them, since that C2852 was closely related to *Stenotrophomonas africana*, C1657 to *Xanthomonas retroflexus*, but C4297, C960, and C2866 were clustered in independent branches, suggesting that these strains could be designated as novel *Stenotrophomonas* species (Fig 4).

## Discussion

Infections caused by *Stenotrophomonas maltophilia* and related species remain challenging to treat due to limited options resulting from both intrinsic and acquired antimicrobial resistance. The introduction of β-lactamase inhibitors has helped to expand the treatment options for certain multidrug-resistant Gram-negative infections. However, it is important to note that currently none of the recently available commercial β-lactamase inhibitors have demonstrated approved activity against *S. maltophilia* [23].

In our analysis, we examined a total of 101 clinical specimens. When comparing our results to similar studies conducted in our country, specifically the work by Cruz-Cordova et al., which focused on an outbreak in a tertiary care hospital in Mexico City, we noted some disparities. Their findings revealed high resistance to SXT (76%) and tetracycline (TET) (80%) in 21 isolated samples [24]. Another study conducted by Velázquez-Acosta et al. analyzed 171 clinical samples of bacteremia and pneumonia in a tertiary-care oncology center. They reported a 5.2% (9/171) resistance rate for SXT, in contrast to our findings, and 20% were resistant to LVX [25]. Additionally, studies conducted in Latin America have shown susceptibility rates for SXT of up to 94.4% and 87.8% for LVX [16].

We found that isolates of *S. maltophilia* obtained from invasive samples remain sufficiently susceptible to first-line antibiotics such as SXT and LVX. This susceptibility has been stable for 10 years. However, between 2015 and 2016 we noticed a decline in LVX susceptibility, which fell to 73%. The reduction in susceptibility was not linked with an outbreak but instead was attributed to numerous factors. We found that we imported two strains from other hospitals because the Instituto Nacional de Rehabilitación Luis Guillermo Ibarra Ibarra is a third level center, some cases came to our institution from the first instance and many other patients were referred from other hospitals. Once the patient arrives, we perform the microbiological protocol to know their colonization status with which the patient arrives. In this way, we may correlate the cases of importation of those residents ones. One of the strains was intermediate to LVX, and the last two were from patients with prolonged antibiotic exposure. These differences in resistance patterns among assorted studies emphasize the importance of monitoring for local resistance rates and understanding regional differences in antimicrobial resistance. Continuous monitoring and individualized approaches to antibiotic selection based on local susceptibility patterns, are essentials, as emphasized in previous studies [26].

Among the antibiotics evaluated *in vitro*, MIN and TGC displayed the highest overall activity. Despite TGC not having approved breakpoints, further investigation is warranted for its potential use in in vivo scenarios. On the other hand, CAZ exhibited a high percentage of resistance (77.22%) and is not recommended as an alternative treatment option. In fact, the Infectious Diseases Society of America (IDSA) guidelines advise against using CAZ as a treatment option, even if it seems susceptible in the antibiogram. This is because of the presence of intrinsic L1 and L2 β-lactamases, which are known to make CAZ ineffective [27]. As expected, clinical isolates that were resistant to CZA were also resistant to CAZ alone. It is worth noting that although CZA significantly decreased the $MIC_{50}$ and $MIC_{90}$ ($p<0.001$), there are no established breakpoints in the CLSI/EUCAST guidelines [28]. In a previous study by Moriceau et al, CZA demonstrated superior results compared to CAZ alone in terms of the proportion of susceptible isolates (66.7% vs. 38.9%, $p<0.01$) and MIC50 (2 μg/mL vs. 12 μg/mL, $p<0.05$) [29]. However, this option still appears appealing and mandates additional scrutiny and assessment.

MEV and IMR showed no activity against *S. maltophilia* strains, indicating their inherent resistance. Therefore, they should not be considered as alternative treatment options [30].

It is noteworthy that none of the clinical isolates that exhibited resistance to any of the ß-lactamase inhibitors had previous drug exposure.

Regarding tetracycline family results, it is noteworthy that all isolates showed susceptibility to MIN and TGC had MICs <4 μg/mL therefore could be used as a potential option in case of multidrug resistance. However, TGC's usage in severe clinical cases is often limited due to rising trends in clinical failure, mortality, and adverse events [31], even though it shows in vitro activity against *S. maltophilia*. Several real-world studies have reported gastrointestinal symptoms as a common adverse event associated with TGC administration [31]. Therefore, it is important to exercise caution when contemplating the use of TGC as a treatment for severe *S. maltophilia* infections, and it is recommended in the IDSA guideline [32] to consider combined options. TGC possesses a lateral lipophilic group that enables it to circumvent the primary resistance mechanisms of second-generation tetracyclines, specifically efflux pumps and ribosomal protection proteins. This property allows for its utility in MIN-resistant *S. maltophilia* infections [33]. Nevertheless, a ventilator-associated pneumonia multicenter retrospective study documented by Zha et al. showed that the clinical outcomes of patients treated with TGC were worse in comparison to that of fluoroquinolones [34]. MIN may serve as a viable option for oral treatment given its superior susceptibility activity, high distribution volume, high-dose regimen (200 mg twice daily) probability of target attainment across MIC distribution [35], and its well-known automated system testability [36].

Five strains, named CRSS (C960, C1657, C2852, C2866, and C4297) showed MICs >2 µg/mL for FDC, which accurately targets *S. maltophilia*. However, the lack of clinical evidence makes it difficult to determine breakpoints [37]. Stracquadanio et al. conducted an in vitro assessment of FDC MICs in 127 clinical isolates. Results showed that FDC had the lowest MIC values and achieved complete efficacy when compared to colistin, CZA, and ceftolozane/tazobactam. This results suggest that FDC may be a highly effective treatment for *S. maltophilia* infections [38]. Recently, the resistance of *S. maltophilia* resistance to FDC has been attributed to various mechanisms such as mutations in the siderophore receptor, mutations affecting porin and efflux pump expression/function, and mutations in the membrane transporter and fimbriae adhesion sites. Therefore, performing genomic sequencing of these strains could be valuable in identifying their resistance mechanism [39].

Since five strains exhibited FDC resistance, we opted to perform WGS of the CRSS for identifying GACR. Previous studies shows that at least 14 components have been associated with FDC resistance [39,40]. Although β-lactam resistance is primarily conferred by the presence of beta lactamases, the presence of mutations in *tonB*, a membrane receptor, may be responsible for conferring FDC resistance to *S. maltophilia* by preventing the active transport of cefiderocol-siderophore conjugates [11,41,42]. Other factors, including iron transporters, type-1 fimbriae, MDR and core metabolic proteins, have been directly or indirectly correlated with FDC resistance in *S. maltophilia* [40,43]. Presumably, mutations in the transcriptional regulator s*meT* are linked to FDC resistance, enabling increased expression of *smeDEF* [40,43]. Mutations in other membrane proteins that allow for the active transport of siderophore conjugates with FDC, such as *tolQ* or transporters like *smlt1418*, as well as Type-1 fimbriae, also have been linked to FDC resistance [39]. Even mutations in elements involved in amino acid and fatty acid metabolism, such as *cysk1*, *panD*, *smlt3318*, and *tesB*, can contribute to FDC resistance [40]. Indeed, we suspect there are several mechanisms that may contribute to FDC resistance in our strains. However, further studies are necessary to clarify the precise mechanism by which *Stenotrophomonas* spp., confer resistance to FDC.

Currently, the species identification based on phenotypic methods is not decisive and lack of accuracy. Therefore, we have decided to perform a presumptive phylogenetic identification of the SRSS. The phylogenetic and pairwise comparisons of 16S-rRNA (>98.5%) suggest that four SRSS can be recognized as members of the Smc species, but the identity percentage for C1657 (<90%) indicates that it may be a new species according to the species designation criteria based on the 16S-rRNA identity percentage [44]. *S. maltophilia*, *Stenotrophomonas septilia*, and *Stenotrophomonas africana* are the only three reported species recognized as human opportunistic pathogens [4,45–47]. Therefore, our research indicates that a greater number of *Stenotrophomonas spp.*, participate in nosocomial infections. During this study, we carefully refrained from assigning a definitive species name to our SRSS. The phylogenetic analysis of 16s-rRNA is insubstantial for species identification for the genus *Stenotrophomonas* [48]. Nonetheless, due comparative identity percentage among species belonging Smc is very similar and maintain high levels 16S gene sequence conservation [4,5], the identification based on this marker is not resolute yet.

Therefore, we plan to conduct a more rigorous taxonomic analysis based on phylogenomic and comparative genomics in a future study to refine and obtain a more species identification.

It is crucial to recognize that our current study has several limitations. First, our study only collected and analyzed data from a single center, potentially limiting the generalization of our findings to other settings. To improve the epidemiological context and achieve a more comprehensive understanding, we suggest establishing a future network dedicated to studying this specific gram-negative rod. Second, we did not evaluate the susceptibility to monobactams. It would be advantageous to incorporate ATM susceptibility testing in upcoming research to

enable us to compare the effectiveness of various inhibitors in the presence of ATM. Lin Q. et al performed a comparable analysis and examined the potency of ATM-Avibactam, showcasing its potential as a treatment option [21]. Conversely, a crucial aspect of our study is that we assessed the activity of novel ß-lactams, such as FDC, and new β-lactamase inhibitors. Assessing the *in vitro* activity of these new beta-lactams and beta-lactamase inhibitors is critical in expanding our arsenal for therapy and overcoming the challenges posed by antimicrobial resistance.

## Conclusion

Resistance to first-line antibiotics SXT, LVX, and MIN is low, making them viable therapeutic alternatives in our institution. TGC and FDC exhibit the most potent in vitro activity compared to the other antibiotics tested, indicating their potential use in future clinical trials. Additionally, exploring the resistance mechanisms of non-susceptible strains to FDC is of interest. AVI restored CAZ's activity and decreased its MIC range by one dilution. This positions it as a viable option for comparison with other antibiotics, including monobactams.

## Material and methods

### Ethics statements

The research committee approved the current protocol assigning the number **2022/054**. No individuals were recruited, animals or cell lines were used; therefore, no ethics committee approval or informed consents were required.

### Strains collection

We included all strains of *S. maltophilia* isolated from invasive specimens, including blood cultures, lower respiratory tract samples, abscesses, and tissue biopsies, over a ten-year period from January 2012 to January 2022. Duplicated samples were excluded, and only recovered isolates were considered for analysis. Sample collection and processing were performed at the Clinical Microbiology Laboratory of the Instituto Nacional de Rehabilitación Luis Guillermo Ibarra Ibarra in Mexico City. All isolates were stored at -70˚C and then inoculated onto sheep blood agar prior to their use. Identification was confirmed through phenotypic characteristics, standard biochemical tests, and the Vitek 2 System (bioMérieux, Marcy ′Étoile, France).

### Minimal inhibitory concentrations determination

The evaluated antibiotics and inhibitors were LVX, SXT, FDC (MedChem Express®, Monmouth Junction, New Jersey, USA), vaborbactam, avibactam, relebactam (Cayman Chemical Company®, Ann Arbor, Michigan, USA), and meropenem (MEM), imipenem (IMP), CAZ, TGC and MIN (Sigma Aldrich®, Saint Louis, Missouri, USA).

Broth microdilution was performed according to the recommendations of the CLSI guidelines [14]. The type strains *Escherichia coli* ATCC 25922, *Klebsiella pneumoniae* ATCC BAA-1705 (KPC producer) and *Enterobacter cloacae* ATCC BAA-2408 (NDM producer) were used for quality control of selected antibiotics. Breakpoints were defined according to the M100 from CLSI 2023 for SXT, LVX, CAZ, FDC and MIN [28]. For agents without a reported MIC, the MIC was defined as the lowest concentration that inhibits bacterial growth. MIC values were reported as $MIC_{50}$, $MIC_{90}$ and MIC range. Results of descriptive analysis were reported in descriptive statistics as percentages and frequencies. To compare the susceptibility of CAZ and ceftazidime/avibactam (CZA), non-parametric statistics were performed with the Wilcoxon sign test for paired samples; a p-value of $\leq 0.05$ was defined as significant.

## Whole Genome Sequencing (WGS)

Five Cefiderocol-Resistant *Stenotrophomonas* strains (CRSS) were chosen for WGS to understand the resistance mechanism associated with FDC. Genomic DNA was extracted using the DNeasy Blood & Tissue (QIAGEN, Germantown, Maryland, USA), according to the manufacturer's instructions. After extraction, DNA quality and quantity were determined using the Qubit 3.0 fluorometer (Invitrogen, Thermo Fisher Scientific Waltham, Massachusetts, USA) and the Nanodrop One/One spectrophotometer (Thermo Fisher Scientific, Waltham, Massachusetts, USA). DNA samples were stored at -20˚C, until use. Library preparation was performed according to the manufacturer's instructions using the Illumina DNA Prep (Illumina, San Diego, California, USA) for the labelling and cleaning steps and IDT for Illumina DNA/RNA UD Indexes (Illumina, San Diego, California, USA) for the indexing step. Library quality control was performed using the Qubit 3.0 fluorometer (Invitrogen, Thermo Fisher Scientific Waltham, Massachusetts, USA) and the 4200 Tapestation System (Agilent, Santa Clara, California, USA). Sequencing of pooled and normalized libraries was performed using the MiSeq Reagent Kit V2 (300 cycles) on the Illumina MiSeq (Illumina, San Diego, California, USA) platform in a paired end configuration. Samples were sequenced at the Centro Nacional de Referencia de Inocuidad y Bioseguridad Agroalimentaria. Servicio Nacional de Sanidad, Inocuidad y Calidad Agroalimentaria.

## Bioinformatic analysis

Quality metrics analysis for the generated WGS raw reads generated was performed using FastQC version 0.11.9 [49]. Raw reads were trimmed using Trimmomatic version 0.39 (parameters: LEADING 3; TRAILING 3; SLIDINGWINDOW 4:20) to remove contaminations, artifacts, and reads with low Phred quality in the raw reads [50]. The genome was assembled with SPAdes version 3.12.0 [51] by using default parameters, combined with the—careful and—cov-cutoff auto options to reduce mismatches errors during assembly. Contigs less than 200 pb in length were removed using setk-seq version 1.3 (https://github.com/lh3/seqtk/). Quality metrics evaluation of assembled contigs was performed using QUAST version 5.2.0 [52]. Encoding nucleotide sequences and their translated amino acid sequences were predicted and annotated using Prokka version 1.14.6 [53].

Genes Associated with Cefiredocol Resistance (GACR) were annotated and identified using ABRIcate version 1.10.1 (https://github.com/tseemann/abricate), using the Comprehensive Antibiotic Resistance Database (CARD) database. Presumptive species assignment and identification was performed by phylogenetic evaluation based on the analysis of the 16S rRNA gene. The 16S gene sequences belonging to *Stenotrophomonas* type strains were downloaded from the DSMZ-German Collection of Microorganisms and Cell Cultures (https://www.dsmz.de/dsmz) [54] and the National Institute for the Biotechnological Information (NCBI) (https://www.ncbi.nlm.nih.gov) [55]. The 16S sequences belonging to CRSS were extracted by using barrnap version 0.9 (https://github.com/tseemann/barrnap). A multiple alignment was performed using MUSCLE v3.8.31 [56]. The maximum-likelihood (ML) phylogenetic tree was constructed by using IQ-TREE v2.1.2 [57], by using the TN + F + I evolutionary model and an ultrafast bootstrap statistical support of the branches with 2000 replicates. The phylogeny was visualized and edited using iTOL v6 [58]. Pairwise comparisons to determine the percentage of identity among 16S gene sequences were calculated using MatGat software [59].

## Supporting information

**S1 Fig. Heatmap of pairwise comparison among 16S rRNA sequences.** Degraded panel on the right side reflects the visual representation of identity percentage from the lowest value

(0% in blue) to the highest value (100% in dark red). Numbers into the boxes represent the numerical value of the identity percentages.
(PPTX)

**S1 Table. Strains included in the study with isolation date and type of microbiological sample.**
(XLSX)

## Acknowledgments

To SENASICA and WHO Collaborating Centre on Antimicrobial Resistance in Foodborne and Environmental Bacteria (MEX-33), especially Mayrén Zamora Nava and Cindy Fabiola Hernández Pérez, and all the sequencing and bioinformatics staff for helping us to sequence our samples.

## Author Contributions

**Conceptualization:** Rafael Franco-Cendejas, Luis Esau Lopez Jacome.

**Formal analysis:** Braulio Josué Méndez-Sotelo, Mónica Delgado-Beltrán, Melissa Hernández-Durán, Claudia Adriana Colín-Castro, José Esquivel-Bautista, Sandra Angélica Ortega-Oliva, Jossue Ortiz-Álvarez, Rodolfo García-Contreras, Rafael Franco-Cendejas, Luis Esau Lopez Jacome.

**Methodology:** Mónica Delgado-Beltrán, Melissa Hernández-Durán, Claudia Adriana Colín-Castro, José Esquivel-Bautista, Sandra Angélica Ortega-Oliva, Jossue Ortiz-Álvarez.

**Supervision:** Rafael Franco-Cendejas, Luis Esau Lopez Jacome.

**Writing – original draft:** Braulio Josué Méndez-Sotelo.

**Writing – review & editing:** Mónica Delgado-Beltrán, Jossue Ortiz-Álvarez, Rodolfo García-Contreras, Rafael Franco-Cendejas, Luis Esau Lopez Jacome.

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
