## [Decision Letter · Decision Letter 0]

28 Dec 2023

PONE-D-23-37653In vitro activity of ceftazidime/avibactam, cefiderocol, meropenem/vaborbactam and imipenem/relebactam against clinical strains of the Stenotrophomonas maltophilia complex.PLOS ONE

Dear Dr. Lopez Jacome,

Thank you for submitting your manuscript to PLOS ONE. After careful consideration, we feel that it has merit but does not fully meet PLOS ONE’s publication criteria as it currently stands. Therefore, we invite you to submit a revised version of the manuscript that addresses the points raised during the review process.

We look forward to receiving your revised manuscript.

Kind regards,

Marwan Osman

Academic Editor

PLOS ONE

2. 1.) For studies reporting research involving human participants, PLOS ONE requires authors to confirm that this specific study was reviewed and approved by an institutional review board (ethics committee) before the study began. Please provide the specific name of the ethics committee/IRB that approved your study, or explain why you did not seek approval in this case.

2.) Please provide additional details regarding participant consent. In the ethics statement in the Methods and online submission information, please ensure that you have specified what type you obtained (for instance, written or verbal, and if verbal, how it was documented and witnessed). If your study included minors, state whether you obtained consent from parents or guardians. If the need for consent was waived by the ethics committee, please include this information.

A clean copy of the edited manuscript (uploaded as the new *manuscript* file)”.

Reviewers' comments:

Reviewer's Responses to Questions

**Comments to the Author**

1. Is the manuscript technically sound, and do the data support the conclusions?

Reviewer #1: Yes

Reviewer #2: Yes

2. Has the statistical analysis been performed appropriately and rigorously? 

Reviewer #1: Yes

Reviewer #2: Yes

3. Have the authors made all data underlying the findings in their manuscript fully available?

Reviewer #1: Yes

Reviewer #2: Yes

4. Is the manuscript presented in an intelligible fashion and written in standard English?

Reviewer #1: Yes

Reviewer #2: Yes

5. Review Comments to the Author

Reviewer #1: In this study, the authors described the in vitro activity of some antibiotics against clinical strains of S.maltophilia isolated in one center in Mexico between 2012 and 2022

The introduction is well written including data from recent and different published studies.

The authors used standard methods for antimicrobial testing and they complete their study by performing WGS for some important strains which were resistant to Cefiderocol. However, there are missing information that could be added to the manuscript (see below recommendation)

In the result part of the study, the authors described their findings by including some tables, graph, and figures. They have showed the antimicrobial susceptibility testing against different antibiotics including Ceftazidime avibactam and Cefiderocol. This is the first study in Mexico that shows some data related to those drugs. Interestingly, the authors found some resistant strains to Cefiderocol, and they performed full genome analysis to elucidate the resistance mechanism as well other factors such as genetic relatedness between the strains.

Finally, the authors discussed the results by performing comparison with some other published studies, as well as by including some limitations

Below here some comments and recommendations to be considered by the authors

L162: Of these, 118 (53.8%) were excluded from the analysis for various reasons

For the authors to clarify the reasons behind the exclusion

Fig 1: I would suggest to remove it as it is not providing additional information

I would rather suggest to use a table with the following data :

Patient Demographic (Age, sex)

Site of isolation

Date of isolation

Patient location ( ex, ICU, CCU)

For the result part, I would recommend to change the way of presentation ( ex-Table 1 and Table 2 to be merged in one table

In Figure 2, the authors presented the trend in sensitivity over the 10 years.

However, it is not clear how many strains have been isolated every year in order to conclude whether those percentage are significant or not

L301: We found that we imported two 301 strains from other hospitals

Could the authors elaborate more on this?

L409: This positions it as a viable option for comparison with other antibiotics, including 410 monobactams

It would be a great added value if the authors would be able to test the combination of Ceftazidime avibactam and Aztreonam and share the findings

Reviewer #2: It will be insightful for the scientific community to deliver these results reporting cifederocol effect with genomic data. Authors started from sample collection to phenotypic identification, then performed MICs following the BMD golden standard method. Sequencing and analysis for selected strains performed to add more value to the study.

6. PLOS authors have the option to publish the peer review history of their article (what does this mean?). If published, this will include your full peer review and any attached files.

Reviewer #1: No

Reviewer #2: No

---

## [Author Response · Author response to Decision Letter 0]

9 Jan 2024

Rebbutal letter

First of all, we would like to thank the reviewers for their time and comments on this paper.

Reviewer 1. L162: Of these, 118 (53.8%) were excluded from the analysis for various reasons. For the authors to clarify the reasons behind the exclusión.

Thank you very much for the observation. We wrote, sic- Of these, 118 (53.8%) were excluded from the analysis for various reasons: two were environmental samples, 32 were swab samples, 14 strains were recovered from catheter tips, one from sonication of prosthetic joint material, four had missing information in electronic medical records, 19 were from urine cultures, and 46 were repeated isolates.- We decided to include only strains related with valuable microbiological samples, since microorganisms isolated from swabs could be colonizers rather than infectious. Same case for environmental samples (from beds, surfaces, etc.), for catheter tips the exclusion reason was because we included those strains isolated from blood cultures (central or peripheric), the case for sonication was similar, we decided to include the strain from biopsy, 4 strains did not have information in the electronical record and we wanted to reduce the selection bias to reduce the overestimation, then those repeated strain from a patient were excluded and we only include one from each patient.

Reviewer 1. Fig 1: I would suggest to remove it as it is not providing additional information

I would rather suggest to use a table with the following data :

Patient Demographic (Age, sex)

Site of isolation

Date of isolation

Patient location ( ex, ICU, CCU)

Thank you very much for the observation and recommendation. At the beginning we though to include the data mentioned above, however, the purpose of the study was centered only in the susceptibility pattern. Since we decided to exclude some strains, the inclusion of these variables is in order to express an epidemiological behavior of our strains. We are interested in further works with Stenotrophomonas spp., that’s why, in this case, we did not include location and demographic data. Nevertheless, as a supplementary table we included strain, date, and site of isolation (Table S1). If you do not have problem, we consider an easy way to follow the number of strains with the algorithm included.

Reviewer 1. For the result part, I would recommend to change the way of presentation ( ex-Table 1 and Table 2 to be merged in one table.

You are right with the observation since both tables are linked, many thanks. The first table is the complete MIC distributions for each antibiotic and second one, which depends on the first, is about the concentrations were the most went down. We tried same to adjust as your recommendation; however, it was difficult to follow due to there were several values (for MICs concentrations). We consider it simpler to avoid getting lost along the reading and not confusing the reader.

Reviewer 1. In Figure 2, the authors presented the trend in sensitivity over the 10 years.

However, it is not clear how many strains have been isolated every year in order to conclude whether those percentage are significant or not

Thank you very much. According to your initial recommendation, in Table S1 are included strains’ isolation dates. 

Reviewer 1. L301: We found that we imported two 301 strains from other hospitals

Could the authors elaborate more on this?

Thank you very much for the observation. We added the following sentence sic- because the Instituto Nacional de Rehabilitación Luis Guillermo Ibarra Ibarra is a third level center, some cases came to our institution from the first instance and many other patients were referred from other hospitals. Once the patient arrives, we perform the microbiological protocol to know their colonization status with which the patient arrives. In this way, we may correlate the cases of importation of those residents ones-

Reviewer 1. L409: This positions it as a viable option for comparison with other antibiotics, including 410 monobactams. 

It would be a great added value if the authors would be able to test the combination of Ceftazidime avibactam and Aztreonam and share the findings

This is a great observation, and a huge opportunity niche as you suggest. As you are thinking, ATM will be interesting since S. maltophilia has a metallo ß-lactamase, unluckily, in this moment we do not have aztreonam salt to perform the experiments, neither more avibactam, we decided to include them, but we did not have enough salt, that´s why we did the observation as limitation of our study. However, for sure, it would be interesting to prove it, in combination, in one side avibactam able to inhibit the cephalosporinase and second, aztreonam evading the action of metallo and then, in combination with ceftazidime. As we say before, we are planning to work with Stenotrophomonas and for sure, we will test ATM/ceftazidime/avibactam and we think to include another inhibitor of metallo ß-lactamases. On the other hand, we did not have in that moment ATM in Mexico, and hence there is not ATM available as treatment.

Reviewer 2. 

We want to thank you very much for your time taken to review our work.

Thank you very much, we really appreciate it.

---

## [Decision Letter · Decision Letter 1]

29 Jan 2024

In vitro activity of ceftazidime/avibactam, cefiderocol, meropenem/vaborbactam and imipenem/relebactam against clinical strains of the Stenotrophomonas maltophilia complex.

PONE-D-23-37653R1

Dear Dr. Lopez Jacome,

We’re pleased to inform you that your manuscript has been judged scientifically suitable for publication and will be formally accepted for publication once it meets all outstanding technical requirements.

Kind regards,

Marwan Osman

Academic Editor

PLOS ONE

Additional Editor Comments (optional):

Reviewers' comments:

Reviewer's Responses to Questions

**Comments to the Author**

1. If the authors have adequately addressed your comments raised in a previous round of review and you feel that this manuscript is now acceptable for publication, you may indicate that here to bypass the “Comments to the Author” section, enter your conflict of interest statement in the “Confidential to Editor” section, and submit your "Accept" recommendation.

Reviewer #1: (No Response)

2. Is the manuscript technically sound, and do the data support the conclusions?

Reviewer #1: (No Response)

3. Has the statistical analysis been performed appropriately and rigorously? 

Reviewer #1: (No Response)

4. Have the authors made all data underlying the findings in their manuscript fully available?

Reviewer #1: (No Response)

5. Is the manuscript presented in an intelligible fashion and written in standard English?

Reviewer #1: (No Response)

6. Review Comments to the Author

Reviewer #1: (No Response)

7. PLOS authors have the option to publish the peer review history of their article (what does this mean?). If published, this will include your full peer review and any attached files.

Reviewer #1: No

---

## [Editor Report · Acceptance letter]

26 Mar 2024

PONE-D-23-37653R1 

PLOS ONE

Dear Dr. Lopez Jacome, 

I'm pleased to inform you that your manuscript has been deemed suitable for publication in PLOS ONE. Congratulations! Your manuscript is now being handed over to our production team.

Kind regards, 

on behalf of

Dr. Marwan Osman 

Academic Editor

PLOS ONE